# Stress and Burnout Related to Electronic Health Record Use among Healthcare Providers during the COVID-19 Pandemic in Saudi Arabia: A Preliminary National Randomized Survey

**DOI:** 10.3390/healthcare9101367

**Published:** 2021-10-14

**Authors:** Jwaher A. Almulhem, Raniah N. Aldekhyyel, Samar Binkheder, Mohamad-Hani Temsah, Amr Jamal

**Affiliations:** 1Medical Informatics and E-learning Unit, Medical Education Department, College of Medicine, King Saud University, Riyadh 11362, Saudi Arabia; raldekhyyel@ksu.edu.sa (R.N.A.); sbinkheder@ksu.edu.sa (S.B.); 2Pediatric Department, College of Medicine, King Saud University, Riyadh 11362, Saudi Arabia; mtemsah@ksu.edu.sa; 3Evidence-Based Health Care & Knowledge Translation Research Chair, King Saud University, Riyadh 11362, Saudi Arabia; amrjamal@ksu.edu.sa; 4Family & Community Medicine Department, College of Medicine, King Saud University, Riyadh 11362, Saudi Arabia

**Keywords:** electronic health record, burnout, COVID-19, healthcare providers, Saudi Arabia

## Abstract

Healthcare providers’ burnout may potentially have a negative impact on patient care. The use of the electronic health record (EHR) increases the burden for healthcare providers (HCPs), particularly during the coronavirus disease-2019 (COVID-19) pandemic. This study assessed the stress and burnout related to the use of EHRs and health information technology (HIT) tools among HCPs during COVID-19 in Saudi Arabia. We used a self-developed survey tool. It consisted of five sections; demographics and professional data, experience using EHR, effects of EHR use, use of EHR and technology tools during COVID-19, and health and wellbeing. The survey link was emailed to a random sample of HCPs registered with a national scientific regulatory body. Univariate, bivariate, and multivariate analyses were performed to measure the association between burnout and study variables. A total of 182 participants completed the survey. 50.5% of participants reported a presence of HIT-related stress, and 40.1% reported a presence of burnout. The variables independently associated with burnout were providing tertiary level of care, working with COVID-19 suspected cases, dissatisfaction with EHRs, and agreement with the statement that using EHRs added frustration to the workday. Further research that explores possible solutions is warranted to minimize burnout among HCPs, especially during infectious outbreaks.

## 1. Introduction

With the global spread of the novel coronavirus disease-2019 (COVID-19), the use of technology to support patient care has seen a rapid rise. Technological tools were used to support patient care during the COVID-19 pandemic crisis through early detection of suspected individuals with the virus using artificial intelligence [1,2], generating big data analytics, reporting real-time data [3], and providing virtual care to patients [4]. The use of electronic health records (EHRs) to support patient care during COVID-19 has also been a focus of many healthcare organizations [5]. The EHR effectively handled COVID-19 by including standardized triage systems, enhancing ordering tools, securing communication tools, automating reporting and analysis, and providing telemedicine consultations, and patient portal services [5].

Despite the advantages of using EHRs in healthcare organizations as a tool to manage the COVID-19 outbreak [5], many healthcare providers (HCPs) reported an increased burden, burnout, and work dissatisfaction related to using EHR systems [6,7]. COVID-19 has led to increased levels of anxiety, stress, and psychological burdens reported by HCPs, especially in the acute care settings and frontlines [1,8,9]. In addition, burnout was reported as a leading factor for many HCPs to leave their jobs [10], and reduce their productivity levels [11], which causes significant disruption to healthcare services [10]. The Agency for Healthcare and Quality (AHRQ) defines burnout as a “long-term stress reaction marked by emotional exhaustion, depersonalization, and a lack of sense of personal accomplishment.” The AHRQ published five reasons that contribute to HCPs’ burnout, one of which is the use of EHRs [12].

In Saudi Arabia, as of 10 August 2021, the total number of cases reported was 534,312, with 8311 deaths [13]. During April 2021, the number of COVID-19 cases increased significantly, indicating a second pandemic wave [14]. Researchers reported that the prevalence of burnout among HCPs in Saudi Arabia during COVID-19 was 75% [15]. In response, the Saudi Commission for Health Specialties (SCFHS); a national regulatory independent professional and scientific body responsible for healthcare practitioners’ training, classification, and registration [16] developed a national support program, called “Daem,” meaning “The Supporter”. The program is designed to provide online interactive guidance services for HCPs who experience professional burnout during COVID-19 [17].

Using EHRs might be a significant factor causing burnout [18] by creating an additional burden on HCPs responsible for performing additional tasks during clinical encounters. Kroth et al. [19] hypothesized that “EHR-associated stress adds to the overall stress and could lead to burnout, which may play a role in the quality of patient care”. New tasks were introduced to the traditional clinical workflow with EHR use, such as medication reconciliation and computerized physician order entry (CPOE) [20]. However, during COVID-19, additional tasks became required from HCPs, such as virtual consultations with patients [5,21], secured messaging with clinical teams and patients [5], and the use of e-prescriptions [21]. Such technology-related tasks increased clinicians’ workload, contributing to an increased risk of HCPs’ burnout and stress [19]. Esmaeilzadeh and Mirzaei specifically studied how the use of EHR features affects HCPs’ burnout among providers who cared for confirmed COVID-19 patients. The researchers concluded that the more that the HCPs were concerned about COVID-19, the higher the odds of burnout [22]. The increased reliance on EHR and other technology-based tools during COVID-19 has led many HCPs to rapidly acquire and practice new skills to provide patient care [4,21,23].

To quantify the association between health information technology (HIT)-related stress and burnout, Gardner et al. in 2019 reported that 26% of 4197 surveyed physicians in the United States (U.S.) reported burnout, and 64.2% agreed that EHR adds to their frustration [18]. Other reported symptoms associated with stress, which were related to the presence of HIT were insufficient time for documentation [18], and excessive time spent on EHR after work hours [18,24,25]. In another similar study conducted by Kroth et al., they reported 43% of surveyed clinicians experienced symptoms of burnout. In addition, the researchers found a significant relationship between the EHR design and physician stress and reported causes of burnout due to how information was presented within the EHR [19]. During COVID-19, Esmaeilzadeh and Mirzaei reported that 36% of HCPs had at least one symptom of burnout. Additional HIT-related stress factors identified were insufficient EHR training, and increased automation systems at work [22].

HCPs’ burnout caused by EHR use can vary across different countries. For example, for the same EHR vendor software “Epic Systems”^®^, physicians in Australia reported satisfaction compared to physicians in the U. S. due to the countries’ variations of regulations and clinical documentation requirements [10]. While the effectiveness of using EHRs was demonstrated in several studies within Saudi Arabia [23,26], little is known about the side effects and burden of using these systems among HCPs during COVID-19 [27].

The importance of measuring and reporting the current state of satisfaction and how it may impact feelings of stress and burnout among HCPs in Saudi Arabia may potentially assist in improving the system design, decrease HCPs’ burnout, and increase satisfaction, especially during the COVID-19 pandemic. Without particular studies that measure the level of HCPs’ burnout from using the EHRs, we are unable to understand the underlying reasons for HCPs’ burnout, improve the system features, and enhance supporting policies. While some studies measured the effective use of other technology tools during COVID-19 [21,23,26], to our knowledge, no study has assessed the burden related to the use of EHRs and other HIT tools during the COVID-19 pandemic among HCPs in our region. Therefore, our aim was to assess stress and burnout related to the use of EHRs and other HIT tools among HCPs during COVID-19 in Saudi Arabia.

## 2. Materials and Methods

### 2.1. Survey Development 

We used a self-developed survey instrument that was constructed based on reviewing several studies related to burnout, EHRs, and COVID-19 [9,18,20,28,29,30]. Before distributing the survey, we conducted a pilot study [31] by asking 37 HCPs, representing different specialties, to review the instrument and test the feasibility of the study. Based on the feedback received, a few wording issues and answer options were modified. The final survey instrument consisted of five sections with a total of 35 questions: (1) demographics and professional data, (2) experience using EHR systems, (3) effects of EHR use, (4) use of EHRs features and technology tools during COVID-19, and (5) health and wellbeing (Appendix A). The questions were developed in several formats: closed-ended, 5-point Likert scale, and binary “yes” and “no” questions. Section five that measured HIT-related stress included three questions (sufficiency of time for documentation at work, the amount of time spent using the EHR remotely at home, and does the use of EHR add frustration) used by Haskell J, et al. [18] to measure HIT-related stress adapted from the Mini z Burnout Survey [28]. Section 5 also measured self-perception of burnout using a single reliable item developed by Schmoldt et al. [29] and validated among HCPs [30], which consisted of a 5-point Likert scale identifying symptoms of burnout among participants. To identify if the burnout resulting from EHR use has increased during the pandemic, we asked the participants about their perceptions of burnout (Appendix A).

### 2.2. Survey Distribution

We used the SCFHS email database to distribute the survey [16]. The survey invitation including the purpose of the study with a link to the web-based survey was emailed to a random sample of 10,000 HCPs registered at the SCFHS. Self-identifying practicing HCPs who use the EHR at their respective workplaces were eligible for the study. Our study excluded interns, medical students, and HCPs who do not use EHRs in their workplace. The survey was open for nine months with reminder emails sent two weeks after the first invitation.

This study was approved by the Institutional Review Board at King Saud University, College of Medicine (IRB# E-3-20-4938).

### 2.3. Statistical Analysis

We used SPSS 19 software to produce descriptive statistics, covering demographic and professional characteristics, HIT characteristics, burnout, technology tools used during COVID-19, and participants’ perception regarding burnout associated with EHR use during COVID-19. For analysis purposes and due to the close response percentages, we combined responses of 5-point Likert scales into three categories [18].

Our two main variables are burnout and HIT- related stress. We determined the presence of HIT-related stress if one or more of the following response categories were indicated: (1) agree/strongly agree that EHRs add to the frustration of the workday, (2) moderately high/excessive use of the EHR at home, and/or (3) poor/marginal time for documentation [18]. To determine burnout among our sample, participants who had <2 on the 5-point scale were defined as “do not have symptoms of burnout,” while participants who had >3 on the 5-point scale were recognized as “having one or more symptoms of burnout” [18].

To measure the association between burnout and demographic, professional, and HIT characteristics among participants, we calculated odds ratio (OR) in unadjusted and adjusted logistic regression models [18,22,32]. Logistic regression is used to measure the association of one or more independent (predictor) variables with a binary dependent (outcome) variable. The strength of the association is measured by reporting OR [33]. The independent variables in our study were demographic, professional, and HIT characteristics, while the dependent variable was presence of burnout. The univariate logistic regression model includes only one predictor, which is reported by the unadjusted OR. The multivariate logistic regression includes more than one predictor variables, which are reported by the adjusted ORs. We only presented the significant variables found in the unadjusted simple logistic regression model for the multivariate regression model. The significance level was assessed at 0.05.

## 3. Results

A total of 182 participants completed the survey. We could not calculate the response rate due to our inability to determine which emails were up-to-date and delivered to the inboxes of HCPs. More than half of the participants were younger than 45 years, male, married, and non-Saudi (Table 1). Most of the participants were physicians working in governmental hospitals providing tertiary care with average reported working hours between 40 to 49 h per week. A total of 72 (40%) participants provided clinical care to COVID-19 suspected patients. The average number of on-calls was 2.34 per week. Participants indicated an average of around 6 years using EHRs. Among our participants, 113 (62.1%) physicians completed the survey, with 26 (14.3%) specialized in internal medicine and 70 (38.5%) were consultants (Table 1).

When asked about the level of satisfaction with EHR, 107 (58.8%) participants were either satisfied or very satisfied. A total of 151 out of 182 (83%) participants indicated that in general they did not have access to the EHR from home prior to the COVID-19 pandemic. Among those with access to the EHR from home, 20 out of 31 (64.5%) participants indicated an opportunity to work from home. Regarding the impact of EHR on workflow, more than 60% of participants either agreed or strongly agreed that EHR systems improve communication, enhance patients care and improve patient safety. Furthermore, more than 50% of the participants felt that EHR systems improve their job satisfaction and clinical workflow (Table 2). Regarding the use of technology during the COVID-19 pandemic, 130 (70%) of participants used e-prescription, followed by the use of messaging communication tools 112 (61.5%). The least electronic feature used by participants was the EHR remote access 59 (32.4%) (Figure 1).

A total of 110 (60%) of participants agreed that using shared computer peripheral devices causes fear in becoming infected with the COVID-19, while 94 (51.6%) of participants slightly or moderately reported that the burnout they experience from the use of EHRs and its associated features has increased due to the COVID-19 pandemic (Figure 2).

In regard to HIT-related stress, 92 (50.5%) of participants reported the presence of HIT-related stress. When participants were asked to determine which measures specifically contributed to HIT-related stress, 62 (34.1%) of participants either agreed or strongly agreed that using the EHR adds frustration to their day. A total of 136 (74.7%) participants reported their proficiency in EHR use as satisfactory or good (Table 2). Due to the low number of physicians among each specialty, we did not report HIT-related stress measures based on physicians’ specialty.

Regarding burnout, 73 (40.1%) of participants reported the presence of burnout, with 37 (50%) identifying as physicians. (Table 3).

In the unadjusted model, the following seven variables were significantly associated with the presence of burnout: female gender, tertiary level of care, physician as a health care profession, working with both suspected and confirmed COVID-19 cases, neither satisfied nor dissatisfied and very dissatisfied or dissatisfied with EHR, participants who think or may think that sharing computer peripherals causes fear in becoming infected with the COVID-19, and agreement about using the EHR adds frustration to daily work (Table 4).

In the adjusted model, which included only variables significantly associated with burnout in the unadjusted model, the variables that remained independently associated with burnout were four variables; tertiary level of care, working with COVID-19 suspected cases, very dissatisfied or dissatisfied with EHR, and agreement with the statement that using the EHR adds frustration to the workday. Working in tertiary healthcare organizations was associated with 5.077 times the odds of burnout compared to working in primary healthcare organizations (AOR 5.077, 95% CI 1.557 to 16.550). Providing care to suspected COVID-19 cases was associated with 4.059 times the odds of burnout compared to providing care to non-COVID-19 cases (AOR 4.059, 95% CI 1.592 to 10.345). Dissatisfaction with EHR was associated with 3.245 times the odds of burnout (AOR 3.245, 95% CI 1.252 to 8.414). Lastly, agreeing that the EHR adds to daily frustration was associated with higher odds of burnout than neither agreement nor disagreement and disagreement with the statement (AOR 5.569, 95% CI 2.159 to 14.366) (Table 4).

## 4. Discussion

Our objective was to measure the presence of stress and burnout among HCPs from using EHRs and HIT tools during the COVID-19 pandemic. It is critical to develop methods to measure and report levels of stress and burnout from using EHR systems on a national level, representing different healthcare sectors in the country. Our findings demonstrated variability in the demographics of our participants and their reported levels of stress and burnout. Our study is relevant to the current global effects of the pandemic on HCPs and their daily use of technology in providing patient care within the Saudi context. Results from our study add to a growing body of evidence calling for a focus on measuring the effects of technology used by HCPs and its impact on stress and burnout [18,22,34,35].

Although overall satisfaction with EHRs among our participants was relatively high, HIT-related stress and burnout reports were found. Half of our participants reported HIT-related stress, with slightly half reporting burnout. These results were consistent with what has been reported previously by other researchers [18,19,22,36]. In addition, more than half moderately reported that the burnout they experience from EHRs and their associated features has increased due to the COVID-19 pandemic. Several predictors were associated with HCPs’ burnout. In particular, participants who provided care to suspected COVID-19 patients, and providing tertiary level of care. Furthermore, our study identified several predictors related to EHR, which contribute to increased odds of burnout; dissatisfaction with EHR, and feeling that EHRs add to daily frustration. Thus, HCPs who were not satisfied with EHRs were more likely to experience burnout compared to HCPs who were satisfied with EHRs. Similarly, HCPs who felt that EHRs add to daily frustration were more likely to experience burnout compared with HCPs who did not report frustration.

During the COVID-19 pandemic, many participants believed that the use of EHRs adds frustration to their daily tasks, which increases their burnout [37]. Most participants reported a higher tendency of using e-prescriptions followed by messaging communication tools such as SMS or WhatsApp. While working on e-prescriptions was found to be time-consuming [38], it is essential to develop EHR tasks, which support clinical processes and automation, such as the use of machine learning and artificial intelligence [38]. Providing EHR customization and one-to-one training sessions are examples of suggested strategies to improve reported burnout among HCPs using EHR tools [22,39,40]. While using the EHR remote access feature was not common among our participants before the pandemic, accessing EHRs remotely has been increased during the pandemic. Indeed, the number of HCPs working from home has increased with the pandemic. Their work from home involved several duties such as providing virtual consultations and clinical advice, conducting virtual triage, and providing virtual prescriptions [41].

Safety concerns about the fear of the spread of COVID-19 were linked to burnout among participants. More than half of the participants were concerned when dealing with shared devices, e.g., computers and tablets, within the hospital during COVID-19. In addition, caring for suspected COVID-19 cases was associated with higher burnout rates among participants than providing care to non-COVID-19 cases. This reflects that participants were concerned about themselves and their families’ safety [36], which was in line with studies reporting fear of being infected was a leading risk factor of anxiety and depression in the workplace [42,43]. These safety concerns need to be addressed to decrease burnout experience among HCPs by providing a safe hospital environment, safety training, and developing up-to-date safety guidelines and protocols [36]. These solutions might help decrease the chances of burnout due to safety concerns but may not necessarily work for all HCPs [43]. As the pandemic evolved, healthcare institutions need to develop an integrated psychological response for HCPs with the occupational and psychological challenge of MERS-CoV outbreaks, especially that anxiety over COVID-19 may increase during the pandemic [44,45,46]. It is vital that healthcare facilities provide more monitoring for burnout and enhance emotional and psychological support for all HCPs [47].

Working in a tertiary healthcare organization was associated with more burnout as compared to working in a primary healthcare organization. This may be due to the type of patients seen at tertiary healthcare organizations, i.e., more specialized, and more complicated cases. Such factors should be considered by tertiary hospitals administration to improve HCPs’ wellbeing. Several interventions to decrease burnout might be used during this pandemic, such as increasing medical resources, recruiting additional HCPs, benefiting from telemedicine, and reducing excessive working hours [48].

The stress and burnout were measurable and prevalent among our participants. Specifically, physicians working with EHRs in our sample reported levels of HIT-related stress, particularly those reporting frustration with the use of EHR. Despite that, previous studies showed variations across medical specialties regarding the presence of HIT-related stress. Our findings may not be able to reflect the actual differences in specialty due to the limited number of participants. It is worth exploring the relationship between specialty and burnout on a larger sample to compare the results with other reports, which measured the levels of stress and burnout among specialists during COVID-19, such as the Medscape National Physician Burnout & Suicide Report 2021 [49].

Future studies to explore the HCPs’ burnout and HIT-related stress during the evolving COVID-19 pandemic are warranted. Other research should also examine strategically possible coping solutions that might reduce burnout and HIT-related stress by focusing on enhancing EHRs satisfaction.

Three issues may affect the generalizability of our study. First, we relied on only one distribution method, SCFHS’s email database, to distribute the survey nationally. Although we used different approaches to increase responses, e.g., sending two reminders to HCPs and allowing responses to be anonymous, the responses to our survey were still low. Distributing the survey was at the peak of the pandemic in Saudi Arabia [13] with a long data collection period, which may also be related to inadequate responses among HCPs. Other methods such as social media announcements need to be employed to increase participation in future research. Second, we used one method to capture the perceived stress and burnout related to the use of EHRs during the COVID-19 pandemic as reported by our participants. Other additional methods for measuring stress and burnout related to the use of EHRs, e.g., conducting an observational study and analyzing actual EHR data, may support our study’s findings. Lastly, most of our participants were physicians. Their experiences may not be generalizable to other HCPs. Focusing on other HCPs, e.g., nurses and pharmacists, would potentially explore other dimensions.

## 5. Conclusions

Although our survey response rate was low, our study demonstrated that HIT-related stress, especially the feeling that the use of EHRs adds frustration to the daily work, was a predictor for burnout among HCPs. Working in tertiary hospitals and working with COVID-19-suspected cases were also predictive factors of burnout. Further research that explores possible solutions is warranted to minimize stress and burnout among HCPs, especially during the infectious disease outbreak.

## Figures and Tables

**Figure 1 healthcare-09-01367-f001:**
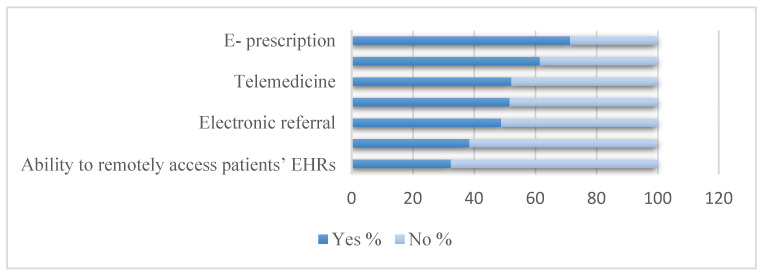
Use of EHR associated features during COVID 19 among participants.

**Figure 2 healthcare-09-01367-f002:**
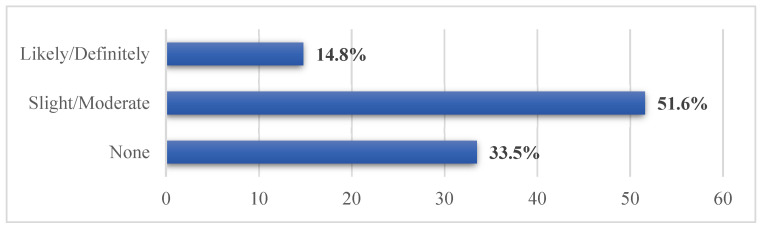
Participants’ perception regarding burnout they experience from the use of EHR and its associated features due to the COVID-19 pandemic.

**Table 1 healthcare-09-01367-t001:** Participants’ demographic and professional characteristics (N = 182).

Characteristic	N (%)
**Age (years)**	
≤45	106 (58.2)
46–65	70 (38.5)
>65	6 (3.3)
**Gender**	
Male	108 (59.3)
Female	74 (40.7)
**Marital status**	
Single	34 (18.7)
Married	141 (77.5)
Widowed or Divorced	7 (3.8)
**Nationality**	
Saudi	82 (45.1)
Non-Saudi	100 (54.1)
**Healthcare organization type**	
Governmental	141 (77.5)
Private	40 (22)
**level of care**	
Primary	35 (19.2)
Secondary	45 (24.7)
Tertiary	102 (56)
**Healthcare profession**	
Physician	113 (62.1)
**Specialty ^1^**	
Anesthesiology	8 (4.4)
Family Medicine	5 (2.7)
Internal Medicine	26 (14.3)
Obstetrics/Gynecology	3 (1.6)
ENT	6 (3.3)
Pediatrics	18 (9.9)
Radiology	8 (4.4)
Surgery	21 (11.5)
Other	18 (9.9)
**Job title ^1,2^**	
Resident	12 (6.6)
Registrar/Senior Registrar	31 (17.0)
Consultant	70 (38.5)
Nurse	44 (24.2)
Pharmacist	9 (4.9)
Other ^3^	16 (8.8)
**Average working hours per week**	
<40	19 (10.4)
40–49	113(62.1)
50–59	23 (12.6)
≥60	27 (14.8)
**Providing care to COVID-19**	
Yes, with COVID-19 suspected	72 (39.6)
Yes, with COVID-19 confirmed	46 (25.3)
No	64 (35.2)
**Number of nights on call per week ^4^**	2.34 (3.351)
**Years of using EHR ^4^**	6.10 (4.422)

^1^ Physicians’ other professional characteristics (N = 113). ^2^ Based on SCFHS categorization. ^3^ Other professions include mostly technicians and dentists. ^4^ Presented as Mean (M) and Standard Division (SD).

**Table 2 healthcare-09-01367-t002:** HIT characteristics among participants (N = 182).

Characteristic	N (%)
**Satisfaction with the EHR system**	
Very dissatisfied/Dissatisfied	40 (22.0)
Neither satisfied nor dissatisfied	35 (19.2)
Satisfied/Very satisfied	107 (58.8)
**EHR remote access from home**	
Yes	31 (17)
No	151 (83)
**Reason for remote EHR use**	
Unable to complete work during regular work hours	3 (9.7)
Have the opportunity to work from home	20 (64.5)
Both the reason	2 (6.5)
Other	6 (19.4)
**Impact of EHR on workflow**	
**EHR improves communication**	
Agree/Strongly agree	112 (61.5)
Neither agree nor disagree	37 (20.3)
Strongly disagree/Disagree	33 (18.1)
**EHR use improves patient care**	
Agree/Strongly agree	115 (63.2)
Neither agree nor disagree	32 (17.6)
Strongly disagree/Disagree	35 (19.2)
**EHR use improves job satisfaction**	
Agree/Strongly agree	108 (59.3)
Neither agree nor disagree	34 (18.7)
Strongly disagree/Disagree	40 (22)
**EHR use improves clinical workflow**	
Agree/Strongly agree	109 (59.9)
Neither agree nor disagree	33 (18.1)
Strongly disagree/Disagree	40 (22.0)
**EHR use improve patient safety**	
Agree/Strongly agree	115 (63.2)
Neither agree nor disagree	38 (20.9)
Strongly disagree/Disagree	29 (15.9)
**HIT-related stress measures**	
**The amount of time spends using the EHR at home**	
Minimal/none	113 (62.1)
Modest/Satisfactory	50 (27.5)
Moderately high/Excessive	19 (10.4)
**Sufficiency of time for documentation at work**	
Poor/Marginal	61 (33.5)
Satisfactory/Good	111 (61)
Optimal	10 (5.5)
**Using the EHR adds frustration to my day**	
Agree/Strongly agree	62 (34.1)
Neither agree nor disagree	52 (28.6)
Strongly disagree/Disagree	68 (37.4)
**Presence of HIT-related stress ^1^**	92 (50.5)
**My proficiency in EHR use**	
Poor/Marginal	22 (12.1)
Satisfactory/Good	136 (74.7)
Optimal	24 (13.2)

^1^ HIT-related stress to be present if one or more of the following response categories were indicated: (1) agree/strongly agree that EHRs add to the frustration of their day, (2) moderately high/excessive use of the EHR at home, and (3) poor/marginal time for documentation.

**Table 3 healthcare-09-01367-t003:** Burnout among participants (N = 182).

Characteristic	N (%)
**Presence of burnout ^1^**	
Yes	73 (40.1)
No	109 (59.9)
**Healthcare profession**	
Physician	37 (50.6)
Nurse	24(32.9)
Pharmacists	7(9.9)
Other	5(6.8)

^1^ Burnout measure was dichotomized into “no presence of burnout” (1 or 2 on a 5-point scale) and “presence of burnout” 3–5 on a 5-point scale).

**Table 4 healthcare-09-01367-t004:** Estimate of the association between demographic, professional, and HIT characteristics and burnout among participants (N = 182).

Characteristic	Unadjusted OR ^1^ (95%CI)	*p*-Value	Adjusted OR ^2^ (95%CI)	*p*-Value
**Gender**				
Male	ref		ref	
Female	2.674 (1.447–4.939)	0.002 ^3^	2.153 (0.936–4.952)	0.071
**Level of care**				
Primary	ref		ref	
Secondary	1.594 (0.602–4.218)	0.348	2.899 (0.804–10.447)	0.104
Tertiary	2.568 (1.095–6.019)	0.030 ^3^	5.077 (1.557–16.550)	0.007 ^3^
**Healthcare profession**				
Physician	0.406 (0.199–0.827)	0.013 ^3^	0.450 (0.163–1.246)	0.124
Nurse	ref		ref	
Pharmacist	2.917 (0.544–15.646)	0.212	9.231 (0.934–91.251)	0.057
Other	0.379 (0.113–1.273)	0.117	0.534 (0.127–2.240)	0.391
**Providing care to COVID-19**				
Yes, with COVID-19 suspected	2.684 (1.292–5.575)	0.008 ^3^	4.059 (1.592–10.345)	0.003 ^3^
Yes, with COVID-19 confirmed	3.000 (1.336–6.736)	0.008 ^3^	2.186 (0.822–5.814)	0.117
No	ref		ref	
**EHR satisfaction**				
Very dissatisfied/Dissatisfied	2.373 (1.130–4.984)	0.022 ^3^	3.245 (1.252–8.414)	0.015 ^3^
Neither satisfied nor dissatisfied	2.273 (1.044–4.948)	0.039 ^3^	0.958 (0.359–2.559)	0.932
Satisfied/Very satisfied	ref		ref	
**Use shared computers**				
Yes	3.692 (1.016–13.418)	0.047 ^3^	2.111 (0.489–9.113)	0.317
No	ref		ref	
Maybe	4.938 (1.283–19.007)	0.020 ^3^	4.270 (0.909–20.054)	0.066
**Using the EHR adds frustration to my day**				
Strongly disagree or Disagree	ref		ref	
Neither agree nor disagree	1.599 (0.733–3.490)	0.238	2.505 (0.944–6.647)	0.065
Strongly agree or Agree	3.846 (1.839–8.045)	0.000 ^3^	5.569 (2.159–14.366)	0.000 ^3^

^1^ Only correlated variables are presented. ^2^ Variables included in adjusted OR are gender, level of care, the healthcare profession, dealing with COVID-19, satisfaction with EHR, use shared computers and using the EHR adds frustration to my day. ^3^
*p* < 0.05.

## Data Availability

Not applicable.

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
