# Peer review of "Stress and Burnout Related to Electronic Health Record Use among Healthcare Providers during the COVID-19 Pandemic in Saudi Arabia: A Preliminary National Randomized Survey"

_healthcare, 2021, doi:10.3390/healthcare9101367_

Round 1

Reviewer 1 Report

Overall:

The research addresses the timely issue of burnout among healthcare providers and concerns about whether electronic health information technology is contributing to the problem. The authors used a self-developed survey tool mailed to a random sample of HCPs with the objective of identifying variables associated with burnout and measuring the association between burnout and study variables. Specifically, the authors aimed to assess stress and burnout related to EHR and other health IT tools among healthcare providers during COVID-19 in Saudi Arabia.

Strengths:

  • The research addresses a timely and important problem.
  • Survey instrument is grounded in prior research, and pilot tested.
  • Overall article is well organized.
  • Methodology seems appropriate although I am not familiar with the specific OR multivariate procedure used.
  • References are relevant and support the data cited.

Issues:

  • Low response rate / generalizability in question. Sample size on which some conclusions are drawn is very small.
  • Would be helpful to reader to explain why OR methodology was selected and why it was deemed most appropriate. Has it been used in other stress/burnout studies? Is cell size for analysis an issue when segmenting populations and drawing conclusions? For example, reporting data by physician specialty with very small numbers (3-26 in each cell). Why report for ObGYN with 3 and not for Internal medicine with 26?)
  • Question related to survey questions 34 and 35: Q34 asks participants to self-report level or burnout without making any reference to EHRs. Q35 then asks participants whether the burnout they are experiencing from EHRs has increased due to COVID-19 pandemic.  Can you explain the relationship of the two questions and how they were interpreted? This was part of my confusion that leads to my next question.
  • Study looks at multiple variables related to EHR burnout. However, relationships are unclear. What is primary cause of stress? EHR vs other variables? 
  • Section 5 Conclusion makes statement that “working tertiary hospital and working with COVID-19 suspected cases are predictive factors of burnout.” Prior discussions talk about associations among variables.  Does the OR methodology support conclusions about predictive associations among variables? More information to amplify this conclusion would be helpful.
  • Study looks at multiple variables related to EHR burnout. However, relationships unclear. What is primary cause of stress? EHR vs other variables?
  • Conclusions—not sure study provides sufficient evidence to support conclusions drawn. I found several questions and inconsistencies in the data and reporting that I was unable to resolve even after spending considerable time.  For example:
    • Only 31/182 participants reported “EHR remote access from home” (Q18). However, results for Q30 (“Amount of time spends using the EHR at Home”—a key measure for assessing burnout) results report 50 modest/satisfactory and 19 Moderately high/Excessive—a total of 69. Can you explain apparent discrepancy?
    • Only 73/182 reported presence of burnout. However, formula to calculate IT related stress/burnout reports 92/182.
    • Lines 56-65. Pre-pandemic, prevalence of burnout among HCP’s was high in region. During pandemic recent study also cites high presence of burnout. The point you are making here is unclear, and how it relates to your other findings.
  • NOTE:  Since I am unfamiliar with the OR procedures used, I was unable to evaluate Table 4.
  • How did you reconcile high satisfaction with EHRs and at the same time EHRs contributing to burnout. This was an unanswered question for me.
  • Some type of diagram or chart to summarize the various relationships among variables used for the study would be helpful. Results are interesting but it was not obvious to me from the data how you reached your conclusions.

Reviewer 2 Report

In this paper, the authors assessed the stress and burnout related to the use of EHR and health information technology (HIT) tools among HCPs during COVID- 19 in Saudi Arabia. A self-developed survey tool is used. The survey link was emailed to a random sample. Univariate, bivariate, and multivariate analysis were performed to measure the association between burnout and study’s variables. From the total of 182 participants, 50.5% of participants reported a presence of HIT-related stress, and 40.1% reported a presence of burnout. It is interesting for possible solutions to minimize burnout among HCPs, especially during the infectious outbreak. The paper is well written and interesting. It would be better if the number of samples could be increased.

Round 2

Reviewer 2 Report

It can be accepted for publish.

Author Response

Dear Reviewer,

Thank you for your comments.